# Constant Nullspace Strong Convexity and Fast Convergence of Proximal Methods under High-Dimensional Settings

**Ian E.H. Yen**      **Cho-Jui Hsieh**      **Pradeep Ravikumar**      **Inderjit Dhillon**

Department of Computer Science
University of Texas at Austin
{ianyen,cjhsieh,pradeepr,inderjit}@cs.utexas.edu

## Abstract

State of the art statistical estimators for high-dimensional problems take the form of regularized, and hence non-smooth, convex programs. A key facet of these statistical estimation problems is that these are typically not strongly convex under a high-dimensional sampling regime when the Hessian matrix becomes rank-deficient. Under vanilla convexity however, proximal optimization methods attain only a sublinear rate. In this paper, we investigate a novel variant of strong convexity, which we call Constant Nullspace Strong Convexity (CNSC), where we require that the objective function be strongly convex only over a constant subspace. As we show, the CNSC condition is naturally satisfied by high-dimensional statistical estimators. We then analyze the behavior of proximal methods under this CNSC condition: we show global linear convergence of Proximal Gradient and local quadratic convergence of Proximal Newton Method, when the regularization function comprising the statistical estimator is decomposable. We corroborate our theory via numerical experiments, and show a qualitative difference in the convergence rates of the proximal algorithms when the loss function does satisfy the CNSC condition.

## 1 Introduction

There has been a growing interest in high-dimensional statistical problems, where the number of parameters $d$ is comparable to or even larger than the sample size $n$, spurred in part by many modern science and engineering applications. It is now well understood that in order to guarantee statistical consistency it is key to impose low-dimensional structure, such as sparsity, or low-rank structure, on the high-dimensional statistical model parameters. A strong line of research has thus developed classes of regularized $M$-estimators that leverage such structural constraints, and come with strong statistical guarantees even under high-dimensional settings [13]. These state of the art regularized $M$-estimators typically take the form of convex non-smooth programs.

A facet of *computational consequence* with these high-dimensional sampling regimes is that these $M$-estimation problems, even when convex, are typically not *strongly convex*. For instance, for the $\ell_1$-regularized least squares estimator (LASSO), the Hessian is rank deficient when $n < d$. In the absence of additional assumptions however, optimization methods to solve general non-smooth non-strongly convex programs can only achieve a sublinear convergence rate [19, 21]; faster rates typically require strong convexity [1, 20]. In the past few years, an effort has thus been made to impose additional assumptions that are stronger than mere convexity, and yet weaker than strong convexity; and proving faster rates of convergence of optimization methods under these assumptions. Typically these assumptions take the form of a restricted variant of strong convexity, which incidentally mirror those assumed for statistical guarantees as well, such as the Restricted Isometry

Property or Restricted Eigenvalue property. A caveat with these results however is that these statistically motivated assumptions need not hold in general, or require sufficiently large number of samples to hold with high probability. Moreover, the standard optimization methods have to be modified in some manner to leverage these assumptions [5, 7, 17]. Another line of research exploits a local error bound to establish asymptotic linear rate of convergence for a special form of non-strongly convex functions [16, 8, 6]. However, these do not provide finite-iteration convergence bounds, due to the potentially large number of iterations spent on early stage.

In this paper, we consider a novel simple condition, which we term Constant Nullspace Strong Convexity (CNSC). This assumption is motivated not from statistical considerations, but from the algebraic form of standard $M$-estimators; indeed as we show, standard $M$-estimation problems even under high-dimensional settings naturally satisfy the CNSC condition. Under this CNSC condition, we then investigate the convergence rates of the class of *proximal optimization methods*; specifically the Proximal Gradient method (Prox-GD) [14, 15, 18] and the Proximal Newton method (Prox-Newton) [1, 2, 9]. These proximal methods are very amenable to regularized $M$-estimation problems: they do not treat the $M$-estimation problem as a black-box convex non-smooth problem, but instead leverage the composite nature of the objective of the form $F(\boldsymbol{x}) = h(\boldsymbol{x}) + f(\boldsymbol{x})$, where $h(\boldsymbol{x})$ is a possibly non-smooth convex function while $f(\boldsymbol{x})$ is a convex smooth function with Lipschitz-continuous gradient. We show that under our CNSC condition, Proximal Gradient achieves global linear convergence when the non-smooth component is a decomposable norm. We also show that Proximal Newton, under the CNSC condition, achieves local quadratic convergence as long as the non-smooth component is Lipschitz-continuous. Note that in the absence of strong convexity, but under no additional assumptions beyond convexity, the proximal methods can only achieve sublinear convergence as noted earlier. We have thus identified an algebraic facet of the $M$-estimators that explains the strong computational performance of standard proximal optimization methods in practical settings in solving high-dimensional statistical estimation problems.

The paper is organized as follows. In Section 2, we define the CNSC condition and introduce the Proximal Gradient and Proximal Newton methods. Then we prove global linear convergence of Prox-GD and local quadratic convergence of Prox-Newton in Section 3 and 4 respectively. In Section 5, we corroborate our theory via experiments on real high-dimensional data set. We will leave all the proof of lemmas to the appendix.

## 2 Preliminaries

We are interested in composite optimization problems of the form

$$\min_{\boldsymbol{x} \in \mathbb{R}^d} \quad F(\boldsymbol{x}) = h(\boldsymbol{x}) + f(\boldsymbol{x}), \tag{1}$$

where $h(\boldsymbol{x})$ is a possibly non-smooth convex function and $f(\boldsymbol{x})$ is twice differentiable convex function with its Hessian matrix $H(\boldsymbol{x}) = \nabla^2 f(\boldsymbol{x})$ satisfying

$$mI \preceq H(\boldsymbol{x}) \preceq MI, \quad \forall \boldsymbol{x} \in \mathbb{R}^d, \tag{2}$$

where for strongly convex $f(\boldsymbol{x})$ we have $m > 0$; otherwise, for convex but not strongly convex $f(\boldsymbol{x})$ we have $m = 0$.

### 2.1 Constant Nullspace Strong Convexity (CNSC)

Before defining our strong convexity variant of Constant Nullspace Strong Convexity (CNSC), we first provide some intuition by considering the following large class of statistical estimation problems in high-dimensional machine learning, where $f(\boldsymbol{x})$ takes the form

$$f(\boldsymbol{x}) = \sum_{i=1}^{n} L(\boldsymbol{a}_i^T \boldsymbol{x}, y_i), \tag{3}$$

where $L(u, y)$ is a non-negative loss function that is convex in its first argument, $\boldsymbol{a}_i$ is the observed *feature vector* and $y_i$ is the observed response of the $i$-th sample. The Hessian matrix of (3) takes the form

$$H(\boldsymbol{x}) = A^T D(A\boldsymbol{x})A, \tag{4}$$

where $A$ is a $n$ by $d$ design (data) matrix with $A_{i,:} = \boldsymbol{a}_i^T$ and $D(A\boldsymbol{x})$ is a diagonal matrix with $D_{ii}(\boldsymbol{x}) = L''(\boldsymbol{a}_i^T\boldsymbol{x}, y_i)$, where the double-derivative in $L''(u, y)$ is with respect to the first argument. It is easy to see that in high-dimensional problems with $d > n$, (4) is not positive definite so that strong convexity would not hold. However, for strictly convex loss function $L(\cdot, y)$, we have $L''(u, y) > 0$ and

$$\boldsymbol{v}^T H(\boldsymbol{x})\boldsymbol{v} = 0 \quad \textit{iff} \quad A\boldsymbol{v} = \mathbf{0}. \tag{5}$$

As a consequence $\boldsymbol{v}^T H(\boldsymbol{x})\boldsymbol{v} > 0$ as long as $\boldsymbol{v}$ does not lie in the Nullspace of $A$; that is, the Hessian $H(\boldsymbol{x})$ might satisfy the strong convexity bound in the above restricted sense. We generalize this concept as follows. We first define the following notation: given a subspace $\mathcal{T}$, we let $\Pi_{\mathcal{T}}(\cdot)$ denote the orthogonal projection onto $\mathcal{T}$, and let $\mathcal{T}^\perp$ denote the orthogonal subspace to $\mathcal{T}$.

**Assumption 1** ( Constant Nullspace Strong Convexity )**.** *A twice-differentiable $f(\boldsymbol{x})$ satisfies Constant Nullspace Strong Convexity (CNSC) with respect to $\mathcal{T}$ (CNSC-$\mathcal{T}$) iff there is a constant vector space $\mathcal{T}$ s.t. $f(\boldsymbol{x})$ depends only on $\boldsymbol{z} = \Pi_{\mathcal{T}}(\boldsymbol{x})$ and its Hessian matrix satisfies*

$$\boldsymbol{v}^T H(\boldsymbol{z})\boldsymbol{v} \geq m\|\boldsymbol{v}\|^2, \quad \forall \boldsymbol{v} \in \mathcal{T} \tag{6}$$

*for some $m > 0$, and $\forall \boldsymbol{z} \in \mathcal{T}$,*

$$H(\boldsymbol{z})\boldsymbol{v} = \mathbf{0}, \quad \forall \boldsymbol{v} \in \mathcal{T}^\perp. \tag{7}$$

From the motivating section above, the above condition can be seen to hold for a wide range of loss functions, such as those arising from linear regression models, as well as generalized linear models (e.g. logistic regression, poisson regression, multinomial regression etc.) [1]. For $L''(u, y) \geq m_L > 0$, we have $m = m_L \lambda_{min}(A^T A) > 0$ as the constant in (6), where $\lambda_{min}(A^T A)$ is the minimum positive eigenvalue of $A^T A$.

Then by the assumption, any point $\boldsymbol{x}$ can be decomposed as $\boldsymbol{x} = \boldsymbol{z} + \boldsymbol{y}$, where $\boldsymbol{z} = \Pi_{\mathcal{T}}(\boldsymbol{x})$, $\boldsymbol{y} = \Pi_{\mathcal{T}^\perp}(\boldsymbol{x})$, so that the difference between gradient of two points can be written as

$$\boldsymbol{g}(\boldsymbol{x}_1) - \boldsymbol{g}(\boldsymbol{x}_2) = \int_0^1 H(s\Delta\boldsymbol{x} + \boldsymbol{x}_2)\Delta\boldsymbol{x}\,ds = \int_0^1 H(s\Delta\boldsymbol{z} + \boldsymbol{z}_2)\Delta\boldsymbol{z}\,ds = \tilde{H}(\boldsymbol{z}_1, \boldsymbol{z}_2)\Delta\boldsymbol{z}, \tag{8}$$

where $\Delta\boldsymbol{x} = \boldsymbol{x}_1 - \boldsymbol{x}_2$, $\Delta\boldsymbol{z} = \boldsymbol{z}_1 - \boldsymbol{z}_2$, and $\tilde{H}(\boldsymbol{z}_1, \boldsymbol{z}_2) = \int_0^1 H(s\Delta\boldsymbol{z} + \boldsymbol{z}_2)\,ds$ is the average Hessian matrix along the path from $\boldsymbol{z}_2$ to $\boldsymbol{z}_1$. It is easy to verify that $\tilde{H}(\boldsymbol{z}_1, \boldsymbol{z}_2)$ satisfies inequalities (2), (6) and equality (7) for all $\boldsymbol{z}_1, \boldsymbol{z}_2 \in \mathcal{T}$ by just applying inequalities (equality) to each individual Hessian matrix being integrated. Then we have following theorem that shows the uniqueness of $\bar{\boldsymbol{z}}$ at optimal.

**Theorem 1** (Optimality Condition)**.** *For $f(\boldsymbol{x})$ satisfying CNSC-$\mathcal{T}$,*

1. *$\bar{\boldsymbol{x}}$ is an optimal solution of (1) iff $-\boldsymbol{g}(\bar{\boldsymbol{x}}) = \bar{\boldsymbol{\rho}}$ for some $\bar{\boldsymbol{\rho}} \in \partial h(\bar{\boldsymbol{x}})$.*

2. *The optimal $\bar{\boldsymbol{\rho}}$ and $\bar{\boldsymbol{z}} = \Pi_{\mathcal{T}}(\bar{\boldsymbol{x}})$ are unique.*

*Proof.* The first statement is true since $\bar{\boldsymbol{x}}$ is an optimal solution iff $\mathbf{0} \in \partial h(\bar{\boldsymbol{x}}) + \nabla f(\bar{\boldsymbol{x}})$. To prove the second statement, suppose $\bar{\boldsymbol{x}}_1 = \bar{\boldsymbol{z}}_1 + \bar{\boldsymbol{y}}_1$ and $\bar{\boldsymbol{x}}_2 = \bar{\boldsymbol{z}}_2 + \bar{\boldsymbol{y}}_2$ are both optimal. Let $\Delta\boldsymbol{x} = \bar{\boldsymbol{x}}_1 - \bar{\boldsymbol{x}}_2$ and $\Delta\boldsymbol{z} = \bar{\boldsymbol{z}}_1 - \bar{\boldsymbol{z}}_2$. Since $h(\boldsymbol{x})$ is convex, $-\boldsymbol{g}(\bar{\boldsymbol{x}}_1) \in \partial h(\bar{\boldsymbol{x}}_1)$ and $-\boldsymbol{g}(\bar{\boldsymbol{x}}_2) \in \partial h(\bar{\boldsymbol{x}}_2)$ should satisfy

$$\langle -\boldsymbol{g}(\bar{\boldsymbol{x}}_1) + \boldsymbol{g}(\bar{\boldsymbol{x}}_2), \Delta\boldsymbol{x} \rangle \geq 0.$$

However, since $f(\boldsymbol{x})$ satisfies CNSC-$\mathcal{T}$, by (8),

$$\langle -\boldsymbol{g}(\bar{\boldsymbol{x}}_1) + \boldsymbol{g}(\bar{\boldsymbol{x}}_2), \Delta\boldsymbol{x} \rangle = \langle -\tilde{H}(\bar{\boldsymbol{z}}_1, \bar{\boldsymbol{z}}_2)\Delta\boldsymbol{z}, \Delta\boldsymbol{x} \rangle = -\Delta\boldsymbol{z}\tilde{H}(\bar{\boldsymbol{z}}_1, \bar{\boldsymbol{z}}_2)\Delta\boldsymbol{z} \leq -m\|\Delta\boldsymbol{z}\|_2^2$$

for some $m > 0$. The two inequalities can simultaneously hold only if $\Delta\bar{\boldsymbol{z}} = \mathbf{0}$. Therefore, $\bar{\boldsymbol{z}}$ is unique at optimum, and thus $\boldsymbol{g}(\bar{\boldsymbol{x}}) = \boldsymbol{g}(\mathbf{0}) + \tilde{H}(\bar{\boldsymbol{z}}, \mathbf{0})\bar{\boldsymbol{z}}$ and $\bar{\boldsymbol{\rho}} = -\boldsymbol{g}(\bar{\boldsymbol{x}})$ are also unique. $\qquad\square$

In next two sections, we review the *Proximal Gradient Method (Prox-GD)* and *Proximal Newton Method (Prox-Newton)*, and introduce some tools that will be used in our analysis.

## 2.2 Proximal Gradient Method

The Prox-GD algorithm comprises a gradient descent step

$$\boldsymbol{x}_{t+\frac{1}{2}} = \boldsymbol{x}_t - \frac{1}{M}\boldsymbol{g}(\boldsymbol{x}_t)$$

followed by a proximal step

$$\boldsymbol{x}_{t+1} = \mathbf{prox}_M^h(\boldsymbol{x}_{t+\frac{1}{2}}) = \arg\min_{\boldsymbol{x}} \ h(\boldsymbol{x}) + \frac{M}{2}\|\boldsymbol{x} - \boldsymbol{x}_{t+\frac{1}{2}}\|_2^2, \tag{9}$$

where $\|\cdot\|_2$ means the Frobinius norm if $\boldsymbol{x}$ is a matrix. For simplicity, we will denote $\mathbf{prox}_M^h(.)$ as $\mathbf{prox}(.)$ in the following discussion when it is clear from the context. In Prox-GD algorithm, it is assumed that (9) can be computed efficiently, which is true for most of decomposable regularizers. Here we introduce some properties of proximal operator that can facilitate our analysis.

**Lemma 1.** *Define* $\Delta^P \boldsymbol{x} = \boldsymbol{x} - \mathbf{prox}(\boldsymbol{x})$, *the following properties hold for proximal operation* (9).

1. $M\Delta^P \boldsymbol{x} \in \partial h(\mathbf{prox}(\boldsymbol{x}))$.

2. $\|\mathbf{prox}(\boldsymbol{x}_1) - \mathbf{prox}(\boldsymbol{x}_2)\|_2^2 \le \|\boldsymbol{x}_1 - \boldsymbol{x}_2\|_2^2 - \|\Delta^P \boldsymbol{x}_1 - \Delta^P \boldsymbol{x}_2\|_2^2$.

## 2.3 Proximal Newton Method

In this section, we introduce the Proximal Newton method, which has been shown to be considerably more efficient than first-order methods in many applications [1], including Sparse Inverse Covariance Estimation [2] and $\ell_1$-regularized Logistic-Regression [9, 10]. Each step of Prox-Newton solves a local quadratic approximation

$$\boldsymbol{x}_t^+ = \arg\min_{\boldsymbol{x}} \ h(\boldsymbol{x}) + \frac{1}{2}(\boldsymbol{x} - \boldsymbol{x}_t)^T H_t(\boldsymbol{x} - \boldsymbol{x}_t) + \boldsymbol{g}_t^T(\boldsymbol{x} - \boldsymbol{x}_t) \tag{10}$$

to find a search direction $\boldsymbol{x}^+ - \boldsymbol{x}_t$, and then conduct a line search procedure to find $t$ such that

$$f(\boldsymbol{x}_{t+1}) = f(\boldsymbol{x}_t + t(\boldsymbol{x}_t^+ - \boldsymbol{x}_t))$$

meets a sufficient decrease condition. Note unlike Prox-GD update (9), in most of cases (10) requires an iterative procedure to solve. For example if $h(\boldsymbol{x})$ is $\ell_1$-norm, then a coordinate descent algorithm is usually employed to solve (10) as an LASSO subproblem [1, 2, 9, 10].

The convergence of Newton-type method comprises two phases [1, 3]. In the first phase, it is possible that step size $t < 1$ is chosen, while in the second phase, which occurs when $\boldsymbol{x}_t$ is close enough to optimum, step size $t = 1$ is always chosen and each step leads to quadratic convergence. In this paper, we focus on the quadratic convergence phase, while refer readers to [21] for a global analysis of Prox-Newton without strong convexity assumption. In the quadratic convergence phase, we have $\boldsymbol{x}_{t+1} = \boldsymbol{x}_t^+$ and the update can be written as

$$\boldsymbol{x}_{t+1} = \mathbf{prox}_{H_t}\left(\boldsymbol{x}_t + \Delta \boldsymbol{x}_t^{nt}\right), \quad H_t \Delta \boldsymbol{x}_t^{nt} = -\boldsymbol{g}_t, \tag{11}$$

where $\Delta \boldsymbol{x}_t^{nt}$ is the Newton step when $h(\boldsymbol{x})$ is absent, and the proximal operator $\mathbf{prox}_H(.)$ is defined for any PSD matrix $H$ as

$$\mathbf{prox}_H(\boldsymbol{x}) = \arg\min_{\boldsymbol{v}} \ h(\boldsymbol{v}) + \frac{1}{2}\|\boldsymbol{v} - \boldsymbol{x}\|_H^2. \tag{12}$$

Note while we use $\|\boldsymbol{x}\|_H^2$ to denote $\boldsymbol{x}^T H \boldsymbol{x}$, we only require $H$ to be PSD instead of PD. Therefore, $\|\boldsymbol{x}\|_H$ is not a true *norm*, and (12) might have multiple solutions, where $\mathbf{prox}_H(\boldsymbol{x})$ refers to any one of them. In the following, we show $\mathbf{prox}_H(.)$ has similar properties as that of $\mathbf{prox}(.)$ in previous section.

**Lemma 2.** *Define* $\Delta^P \boldsymbol{x} = \boldsymbol{x} - \mathbf{prox}_H(\boldsymbol{x})$, *the following properties hold for the proximal operator:*

1. $H\Delta^P \boldsymbol{x} \in \partial h(\mathbf{prox}_H(\boldsymbol{x}))$.

2. $\|\mathbf{prox}_H(\boldsymbol{x}_1) - \mathbf{prox}_H(\boldsymbol{x}_2)\|_H^2 \le \|\boldsymbol{x}_1 - \boldsymbol{x}_2\|_H^2$.

# 3 Linear Convergence of Proximal Gradient Method

In this section, we analyze convergence of Proximal Gradient Method for $h(\boldsymbol{x}) = \lambda\|\boldsymbol{x}\|$, where $\|\cdot\|$ is a decomposable norm defined as follows.

**Definition 1** (Decomposable Norm). $\|\cdot\|$ *is a decomposable norm if there are orthogonal subspaces* $\{\mathcal{M}_i\}_{i=1}^J$ *with* $\mathbb{R}^d = \cup_{i=1}^J \mathcal{M}_i$ *such that for any point* $\boldsymbol{x} \in \mathbb{R}^d$ *that can be written as* $\boldsymbol{x} = \sum_{j \in \mathcal{E}} c_j \boldsymbol{a}_j$, *where* $c_j > 0$ *and* $\boldsymbol{a}_j \in \mathcal{M}_j$, $\|\boldsymbol{a}_j\|_* = 1$, *we have*

$$\|\boldsymbol{x}\| = \sum_{j \in \mathcal{E}} c_j, \quad and \quad \partial\|\boldsymbol{x}\| = \{\boldsymbol{\rho} \mid \Pi_{\mathcal{M}_j}(\boldsymbol{\rho}) = \boldsymbol{a}_j, \forall j \in \mathcal{E}; \ \|\Pi_{\mathcal{M}_j}(\boldsymbol{\rho})\|_* \leq 1, \forall j \notin \mathcal{E}\}, \quad (13)$$

*where* $\|\cdot\|_*$ *is the dual norm of* $\|\cdot\|$.

The above definition includes several well-known examples such as $\ell_1$-norm $\|\boldsymbol{x}\|_1$ and group-$\ell_1$ norm $\|X\|_{1,2}$. For $\ell_1$-norm, $\mathcal{M}_j$ corresponds to vectors with only $j$-th coordinate not equal to 0, and $\mathcal{E}$ is the set of non-zero coordinates of $\boldsymbol{x}$. For group-$\ell_1$ norm, $\mathcal{M}_j$ corresponds to vectors with only $j$-th group not equal to $\mathbf{0}^T$ and $\mathcal{E}$ are the set of non-zero groups of $X$. Under the definition, we can profile the set of optimal solutions as follows.

**Lemma 3** (Optimal Set). *Let* $\bar{\mathcal{E}}$ *be the active set at optimal and* $\bar{\mathcal{E}}^+ = \{j \mid \|\Pi_{\mathcal{M}_j}(\bar{\boldsymbol{\rho}})\|_* = \lambda\}$ *be its augmented set (which is unique since* $\bar{\boldsymbol{\rho}}$ *is unique) such that* $\Pi_{\mathcal{M}_j}(\bar{\boldsymbol{\rho}}) = \lambda\bar{\boldsymbol{a}}_j$, $j \in \bar{\mathcal{E}}^+$. *The optimal solutions of* (1) *form a polyhedral set*

$$\bar{\mathcal{X}} = \left\{\boldsymbol{x} \mid \Pi_{\mathcal{T}}(\boldsymbol{x}) = \bar{\boldsymbol{z}} \text{ and } \boldsymbol{x} \in \bar{\mathcal{O}}\right\}, \quad (14)$$

*where* $\bar{\mathcal{O}} = \left\{\boldsymbol{x} \mid \boldsymbol{x} = \sum_{j \in \bar{\mathcal{E}}^+} c_j \bar{\boldsymbol{a}}_j, c_j \geq 0, j \in \bar{\mathcal{E}}^+\right\}$ *is the set of* $\boldsymbol{x}$ *with* $\bar{\boldsymbol{\rho}} \in \partial h(\boldsymbol{x})$.

Given the optimal set is a polyhedron, we can then employ the following lemma to bound the distance of an iterate $\boldsymbol{x}_t$ to the optimal set $\bar{\mathcal{X}}$.

**Lemma 4** (Hoffman's bound). *Consider a polyhedral set* $\mathcal{S} = \{\boldsymbol{x} \mid A\boldsymbol{x} \leq b, E\boldsymbol{x} = c\}$. *For any point* $\boldsymbol{x} \in \mathbb{R}^d$, *there is a* $\bar{\boldsymbol{x}} \in \mathcal{S}$ *such that*

$$\|\boldsymbol{x} - \bar{\boldsymbol{x}}\|_2 \leq \theta(\mathcal{S}) \left\| \begin{bmatrix} [A\boldsymbol{x} - b]^+ \\ E\boldsymbol{x} - c \end{bmatrix} \right\|_2, \quad (15)$$

*where* $\theta(\mathcal{S})$ *is a positive constant that depends only on* $A$ *and* $E$.

The above bound first appears in [11], and was employed in [4] to prove linear convergence of Feasible Descent method for a class of convex smooth function. A proof of the $\ell_2$-norm version (15) can be found in [4, lemma 4.3]. By applying (15) to the set $\bar{\mathcal{X}}$, the distance of a point $\boldsymbol{x}$ to $\bar{\mathcal{X}}$ can be bounded by infeasible amounts to the two constraints $\Pi_{\mathcal{T}}(\boldsymbol{x}) = \boldsymbol{z}$ and $\boldsymbol{x} \in \bar{\mathcal{O}}$, where the latter can be bounded according the following lemma when $c_j = \langle \boldsymbol{x}, \bar{\boldsymbol{a}}_j \rangle \geq 0, \forall j \in \bar{\mathcal{E}}^+$.

**Lemma 5.** *Let* $\bar{\mathcal{A}} = span(\bar{\boldsymbol{a}}_1, \bar{\boldsymbol{a}}_2 \ldots, \bar{\boldsymbol{a}}_{|\bar{\mathcal{E}}^+|})$. *Suppose* $\|\boldsymbol{x}\| \leq R$ *and* $\Pi_{\mathcal{M}_j}(\boldsymbol{x}) = \mathbf{0}$ *for* $j \notin \bar{\mathcal{E}}^+$. *Then*

$$\lambda^2 \|\boldsymbol{x} - \Pi_{\bar{\mathcal{A}}}(\boldsymbol{x})\|_2^2 \leq R^2 \|\boldsymbol{\rho} - \bar{\boldsymbol{\rho}}\|_2^2,$$

*where* $\boldsymbol{\rho} \in \partial h(\boldsymbol{x})$ *and* $\bar{\boldsymbol{\rho}}$ *is as defined in Theorem 1.*

Now we are ready to prove the main theorem of this section.

**Theorem 2** (Linear Convergence of Prox-GD). *Let* $\bar{\mathcal{X}}$ *be the set of optimal solutions for problem* (1), *and* $\bar{\boldsymbol{x}} = \Pi_{\bar{\mathcal{X}}}(\boldsymbol{x})$ *be the solution closest to* $\boldsymbol{x}$. *Denote* $d_\lambda = \min_{j \notin \bar{\mathcal{E}}^+} (\lambda - \|\Pi_{\mathcal{M}_j}(\bar{\boldsymbol{\rho}})\|_*) > 0$. *For the sequence* $\{\boldsymbol{x}_t\}_{t=0}^\infty$ *produced by Proximal Gradient Method, we have:*

(a) *If* $x_{t+1}$ *satisfies the condition that*

$$\exists j \notin \bar{\mathcal{E}}^+ : \ \Pi_{\mathcal{M}_j}(\boldsymbol{x}_{t+1}) \neq \mathbf{0} \text{ or } \exists j \in \bar{\mathcal{E}}^+ : \ \langle \boldsymbol{x}_{t+1}, \bar{\boldsymbol{a}}_j \rangle < 0, \quad (16)$$

*we then have:*

$$\|\boldsymbol{x}_{t+1} - \bar{\boldsymbol{x}}_{t+1}\|_2^2 \leq (1 - \alpha)\|\boldsymbol{x}_t - \bar{\boldsymbol{x}}_t\|_2^2, \quad \alpha = \frac{d_\lambda^2}{M^2 \|\boldsymbol{x}_0 - \bar{\boldsymbol{x}}_0\|_2^2} \quad (17)$$

*(b) If $x_{t+1}$ does not satisfy the condition in (16) but $x_t$ does, then*

$$\|\boldsymbol{x}_{t+1} - \bar{\boldsymbol{x}}_{t+1}\|_2^2 \le (1-\alpha)\|\boldsymbol{x}_{t-1} - \bar{\boldsymbol{x}}_{t-1}\|_2^2, \quad \alpha = \frac{d_\lambda^2}{M^2\|\boldsymbol{x}_0 - \bar{\boldsymbol{x}}_0\|_2^2} \qquad (18)$$

*(c) If neither $x_{t+1}, x_t$ satisfy the condition in (16), then*

$$\|\boldsymbol{x}_{t+2} - \bar{\boldsymbol{x}}_{t+2}\|_2^2 \le \frac{1}{1+\beta}\|\boldsymbol{x}_t - \bar{\boldsymbol{x}}_t\|_2^2, \quad \beta = \frac{m}{M\theta(\bar{\mathcal{X}})^2}, \qquad (19)$$

*where we recall that $\theta(\bar{\mathcal{X}})$ is the constant determined by polyhedron $\bar{\mathcal{X}}$ from Hoffman's Bound (15).*

*Proof.* Since $\bar{\boldsymbol{x}}_t$ is an optimal solution, we have $\bar{\boldsymbol{x}}_t = \mathbf{prox}(\bar{\boldsymbol{x}}_t - \boldsymbol{g}(\bar{\boldsymbol{x}}_t)/M)$. Let $\Delta\boldsymbol{x}_t = \boldsymbol{x}_t - \bar{\boldsymbol{x}}_t$, $\boldsymbol{\rho}_t = M(\boldsymbol{x}_{t+\frac{1}{2}} - \boldsymbol{x}_{t+1}) \in \partial h(\boldsymbol{x}_{t+1})$ and $\tilde{H} = \tilde{H}(\boldsymbol{z}_t, \bar{\boldsymbol{z}}_t)$. by Lemma 1, each iterate of Prox-GD has

$$\begin{aligned}
\|\boldsymbol{x}_t - \bar{\boldsymbol{x}}_t\|_2^2 - \|\boldsymbol{x}_{t+1} - \bar{\boldsymbol{x}}_{t+1}\|_2^2 &\ge \|\boldsymbol{x}_t - \bar{\boldsymbol{x}}_t\|_2^2 - \|\boldsymbol{x}_{t+1} - \bar{\boldsymbol{x}}_t\|_2^2 \\
&= \|\Delta\boldsymbol{x}_t\|_2^2 - \|\mathbf{prox}(\boldsymbol{x}_t - \boldsymbol{g}(\boldsymbol{x}_t)/M) - \mathbf{prox}(\bar{\boldsymbol{x}}_t - \boldsymbol{g}(\bar{\boldsymbol{x}}_t)/M)\|_2^2 \\
&\ge \|\Delta\boldsymbol{x}_t\|_2^2 - \|(\boldsymbol{x}_t - \boldsymbol{g}(\boldsymbol{x}_t)/M) - (\bar{\boldsymbol{x}}_t - \boldsymbol{g}(\bar{\boldsymbol{x}}_t)/M)\|_2^2 + \|\boldsymbol{\rho}_t - \bar{\boldsymbol{\rho}}\|_2^2/M^2.
\end{aligned} \qquad (20)$$

Since $\boldsymbol{g}(\boldsymbol{x}_t) - \boldsymbol{g}(\bar{\boldsymbol{x}}_t) = \tilde{H}\Delta\boldsymbol{x}$ from (8), we have

$$\begin{aligned}
\|\boldsymbol{x}_t - \bar{\boldsymbol{x}}_t\|_2^2 - \|\boldsymbol{x}_{t+1} - \bar{\boldsymbol{x}}_{t+1}\|_2^2 &\ge \|\Delta\boldsymbol{x}_t\|_2^2 - \|\Delta\boldsymbol{x}_t - \tilde{H}\Delta\boldsymbol{x}_t/M\|_2^2 + \|\boldsymbol{\rho}_t - \bar{\boldsymbol{\rho}}\|_2^2/M^2 \\
&\ge \Delta\boldsymbol{x}_t^T\left(\tilde{H}/M\right)\Delta\boldsymbol{x}_t + \|\boldsymbol{\rho}_t - \bar{\boldsymbol{\rho}}\|_2^2/M^2 \\
&\ge m\|\Delta\boldsymbol{z}_t\|_2^2/M + \|\boldsymbol{\rho}_t - \bar{\boldsymbol{\rho}}\|_2^2/M^2.
\end{aligned} \qquad (21)$$

The second inequality holds since $2\tilde{H}/M - \tilde{H}^2/M^2 = (\tilde{H}/M)(2I - \tilde{H}/M) \succeq \tilde{H}/M$. The inequality tells us $\|\boldsymbol{x}_t - \bar{\boldsymbol{x}}_t\|^2 - \|\boldsymbol{x}_{t+1} - \bar{\boldsymbol{x}}_{t+1}\|^2 \ge 0$, that is, the distance to the optimal set $\|\boldsymbol{x}_t - \bar{\boldsymbol{x}}_t\|$ is monotonically non-increasing. To get a tighter bound, we consider two cases.

**Case 1:** $\Pi_{\mathcal{M}_j}(\boldsymbol{x}_t) \ne \mathbf{0}$ for some $j \notin \bar{\mathcal{E}}^+$ or $\langle\boldsymbol{x}_t, \bar{\boldsymbol{a}}_j\rangle < 0$ for some $j \in \bar{\mathcal{E}}^+$.

In this case, suppose there is $j \notin \mathcal{E}_t^+$ with $\Pi_{\mathcal{M}_j}(\boldsymbol{x}_t) \ne \mathbf{0}$, then [2]

$$\|\boldsymbol{\rho}_t - \bar{\boldsymbol{\rho}}\|_2^2 \ge \|\Pi_{\mathcal{M}_j}(\boldsymbol{\rho}_t) - \Pi_{\mathcal{M}_j}(\bar{\boldsymbol{\rho}})\|_*^2 \ge (\|\Pi_{\mathcal{M}_j}(\boldsymbol{\rho}_t)\|_* - \|\Pi_{\mathcal{M}_j}(\bar{\boldsymbol{\rho}})\|_*)^2 \ge d_\lambda^2. \qquad (22)$$

On the other hand, if $\langle\boldsymbol{x}_t, \bar{\boldsymbol{a}}_j\rangle < 0$ for some $j \in \bar{\mathcal{E}}^+$, then we have $\langle\boldsymbol{a}_j, \bar{\boldsymbol{a}}_j\rangle < 0$ for $\Pi_{\mathcal{M}_j}(\boldsymbol{\rho}_t) = \lambda\boldsymbol{a}_j$. Therefore

$$\|\boldsymbol{\rho}_t - \bar{\boldsymbol{\rho}}\|_2^2 \ge \|\Pi_{\mathcal{M}_j}(\boldsymbol{\rho}_t) - \Pi_{\mathcal{M}_j}(\bar{\boldsymbol{\rho}})\|_2^2 \ge \lambda^2\|\boldsymbol{a}_j - \bar{\boldsymbol{a}}_j\|_2^2 = \lambda^2(2 - 2\langle\boldsymbol{a}_j, \bar{\boldsymbol{a}}_j\rangle) > 2\lambda^2.$$

Either cases we have

$$\|\boldsymbol{x}_t - \bar{\boldsymbol{x}}_t\|_2^2 - \|\boldsymbol{x}_{t+1} - \bar{\boldsymbol{x}}_{t+1}\|_2^2 \ge \frac{\|\boldsymbol{\rho}_t - \bar{\boldsymbol{\rho}}\|_2^2}{M^2} \ge \left(\frac{d_\lambda^2}{M^2\|\boldsymbol{x}_0 - \bar{\boldsymbol{x}}_0\|_2^2}\right)\|\boldsymbol{x}_t - \bar{\boldsymbol{x}}_t\|_2^2. \qquad (23)$$

**Case 2:** Both $\boldsymbol{x}_t, \boldsymbol{x}_{t+1}$ do not fall in **Case 1**

Given $\langle\boldsymbol{x}_t, \bar{\boldsymbol{a}}_j\rangle \ge 0, \forall j \in \bar{\mathcal{E}}^+$ and $\Pi_{\mathcal{M}_j}(\boldsymbol{x}_t) = \mathbf{0}, \forall j \notin \bar{\mathcal{E}}^+$, then $\boldsymbol{x}$ belongs to the set $\bar{\mathcal{O}}$ defined in Lemma 3 iff $\|\boldsymbol{x} - \Pi_{\bar{\mathcal{A}}}(\boldsymbol{x})\|_2^2 = 0$. The condition can be also scaled as $\frac{\lambda^2}{mMR^2}\|\boldsymbol{x} - \Pi_{\bar{\mathcal{A}}}(\boldsymbol{x})\|_2^2 = 0$, where $R$ is a bound on $\|\boldsymbol{x}_t\|$ holds for $\forall t$, which must exist as long as the regularization parameter $\lambda > 0$ in $h(\boldsymbol{x}) = \lambda\|x\|$.

By Lemma 4, the distance of point $\boldsymbol{x}_t$ to the polyhedral set $\bar{\mathcal{X}}$ is bounded by its infeasible amount

$$\|\boldsymbol{x}_t - \bar{\boldsymbol{x}}_t\|_2^2 \le \theta(\bar{\mathcal{X}})^2\left(\|\boldsymbol{z}_t - \bar{\boldsymbol{z}}\|_2^2 + \frac{\lambda^2}{mMR^2}\|\boldsymbol{x}_t - \Pi_{\bar{\mathcal{A}}}(\boldsymbol{x}_t)\|_2^2\right), \qquad (24)$$

where $\boldsymbol{z}_t = \Pi_{\mathcal{T}}(\boldsymbol{x}_t)$. Applying (24) to (21) for iteration $t + 1$, we have

$$\|\boldsymbol{x}_{t+1} - \bar{\boldsymbol{x}}_{t+1}\|^2 - \|\boldsymbol{x}_{t+2} - \bar{\boldsymbol{x}}_{t+2}\|^2$$
$$\geq \frac{m}{M\theta(\bar{\mathcal{X}})^2}\|\Delta\boldsymbol{x}_{t+1}\|^2 - \frac{\lambda^2}{M^2 R^2}\|\boldsymbol{x}_{t+1} - \Pi_{\bar{A}}(\boldsymbol{x}_{t+1})\|_2^2 + \frac{\|\boldsymbol{\rho}_{t+1} - \bar{\boldsymbol{\rho}}\|^2}{M^2}.$$

For iteration $t$, we have

$$\|\boldsymbol{x}_t - \bar{\boldsymbol{x}}_t\|^2 - \|\boldsymbol{x}_{t+1} - \bar{\boldsymbol{x}}_{t+1}\|^2 \geq \frac{m}{M}\|\Delta\boldsymbol{z}_t\|_2^2 + \frac{\|\boldsymbol{\rho}_t - \bar{\boldsymbol{\rho}}\|^2}{M^2}$$

. By Lemma 5, adding the two inequalities gives

$$\|\boldsymbol{x}_t - \bar{\boldsymbol{x}}_t\|^2 - \|\boldsymbol{x}_{t+2} - \bar{\boldsymbol{x}}_{t+2}\|^2 \geq \frac{m}{M\theta(\bar{\mathcal{X}})^2}\|\Delta\boldsymbol{x}_{t+1}\|^2 + \frac{m}{M}\|\Delta\boldsymbol{z}_t\|_2^2 + \frac{\|\boldsymbol{\rho}_{t+1} - \bar{\boldsymbol{\rho}}\|^2}{M^2}$$
$$\geq \frac{m}{M\theta(\bar{\mathcal{X}})^2}\|\Delta\boldsymbol{x}_{t+1}\|^2 \geq \frac{m}{M\theta(\bar{\mathcal{X}})^2}\|\Delta\boldsymbol{x}_{t+2}\|^2,$$

which yields desired result (18) after arrangement. $\square$

We note that the descent in the first two cases is actually even stronger than stated above: from the proofs, that the distance can be seen to reduce by a fixed constant. This is faster than superlinear convergence since the final solution could then be obtained in a finite number of steps.

## 4 Quadratic Convergence of Proximal Newton Method

The key idea of the proof is to re-formulate Prox-Newton update (10) as

$$\boldsymbol{z}_{t+1} = \underset{\boldsymbol{z} \in \mathcal{T}}{\arg\min} \quad h(\boldsymbol{z} + \hat{\boldsymbol{y}}(\boldsymbol{z})) + \boldsymbol{g}_t^T(\boldsymbol{z} - \boldsymbol{z}_t) + \frac{1}{2}\|\boldsymbol{z} - \boldsymbol{z}_t\|_{H_t}^2 \tag{25}$$

where

$$\hat{\boldsymbol{y}}(\boldsymbol{z}) = \underset{\boldsymbol{y} \in \mathcal{T}^\perp}{\arg\min} \quad h(\boldsymbol{z} + \boldsymbol{y}), \tag{26}$$

so that we can focus our convergence analysis on $\boldsymbol{z} = \Pi_{\mathcal{T}}(\boldsymbol{x})$ as follows.

**Lemma 6** (Optimality Condition). *For any matrix $H$ satisfying CNSC-$\mathcal{T}$, the update*

$$\Delta\boldsymbol{x} = \underset{\boldsymbol{d}}{\arg\min} \quad h(\boldsymbol{x} + \boldsymbol{d}) + \boldsymbol{g}(\boldsymbol{x})^T\boldsymbol{d} + \frac{1}{2}\|\boldsymbol{d}\|_H^2 \tag{27}$$

*has*

$$F(\boldsymbol{x} + t\Delta\boldsymbol{x}) - F(\boldsymbol{x}) \leq -t\|\Delta\boldsymbol{z}\|_H^2 + O(t^2), \tag{28}$$

*where $\Delta\boldsymbol{z} = \Pi_{\mathcal{T}}(\Delta\boldsymbol{x})$. Furthermore, if $\boldsymbol{x}$ is an optimal solution, $\Delta\boldsymbol{x} = \boldsymbol{0}$ satisfies (27).*

The following lemma then states that, for Prox-Newton, the function suboptimality is bounded by only distance in the $\mathcal{T}$ space.

**Lemma 7.** *Suppose $h(\boldsymbol{x})$ and $f(\boldsymbol{x})$ are Lipschitz-continuous with Lipschitz constants $L_h$ and $L_f$. In quadratic convergence phase (defined in Theorem 3), Proximal Newton Method has*

$$F(\boldsymbol{x}_t) - F(\bar{\boldsymbol{x}}) \leq L\|\boldsymbol{z}_t - \bar{\boldsymbol{z}}\|, \tag{29}$$

*where $L = \max\{L_h, L_f\}$ and $\boldsymbol{z}_t = \Pi_{\mathcal{T}}(\boldsymbol{x}_t)$, $\bar{\boldsymbol{z}} = \Pi_{\mathcal{T}}(\bar{\boldsymbol{x}})$.*

By the above lemma, we have $F(\boldsymbol{x}_t) - F(\bar{\boldsymbol{x}}) \leq L\epsilon$ as long as $\|\boldsymbol{z}_t - \bar{\boldsymbol{z}}\| \leq \epsilon$. Therefore, it suffices to show quadratic convergence of $\|\boldsymbol{z}_t - \bar{\boldsymbol{z}}\|$ to guarantee $F(\boldsymbol{x}_t) - F(\bar{\boldsymbol{x}})$ double its precision after each iteration.

**Theorem 3** (Quadratic Convergence of Prox-Newton). *For $f(\boldsymbol{x})$ satisfying CNSC-$\mathcal{T}$ with Lipschitz-continuous second derivative $\nabla^2 f(\boldsymbol{x})$, the Proximal Newton update (10) has*

$$\|\boldsymbol{z}_{t+1} - \bar{\boldsymbol{z}}\| \leq \frac{L_H}{2m}\|\boldsymbol{z}_t - \bar{\boldsymbol{z}}\|^2,$$

*where $\bar{\boldsymbol{z}} = \Pi_{\mathcal{T}}(\bar{\boldsymbol{x}})$, $\boldsymbol{z}_t = \Pi_{\mathcal{T}}(\boldsymbol{x}_t)$, and $L_H$ is the Lipschitz constant for $\nabla^2 f(\boldsymbol{x})$.*

*Proof.* Let $\bar{\boldsymbol{x}}$ be an optimal solution of (1). By Lemma 6, for any PSD matrix $H$ the update $\Delta\bar{\boldsymbol{x}} = \mathbf{0}$ satisfies (27), which means

$$\bar{\boldsymbol{x}} = \mathbf{prox}_{H_t}(\bar{\boldsymbol{x}} + \Delta\bar{\boldsymbol{x}}^{nt}), \quad H_t\Delta\bar{\boldsymbol{x}}^{nt} = -\boldsymbol{g}(\bar{\boldsymbol{x}}). \tag{30}$$

Then by non-expansiveness of proximal operation (Lemma 2), we have

$$\begin{aligned}
\|\boldsymbol{x}_{t+1} - \bar{\boldsymbol{x}}\|_{H_t} &= \|\mathbf{prox}_{H_t}(\boldsymbol{x}_t + \Delta\boldsymbol{x}_t^{nt}) - \mathbf{prox}_{H_t}(\bar{\boldsymbol{x}} + \Delta\bar{\boldsymbol{x}}^{nt})\|_{H_t} \\
&\leq \|(\boldsymbol{x}_t + \Delta\boldsymbol{x}_t^{nt}) - (\bar{\boldsymbol{x}} + \Delta\bar{\boldsymbol{x}}^{nt})\|_{H_t} = \|(\boldsymbol{x}_t - \bar{\boldsymbol{x}}) + (\Delta\boldsymbol{x}_t^{nt} - \Delta\bar{\boldsymbol{x}}^{nt})\|_{H_t} \quad (31) \\
&= \|(\boldsymbol{z}_t - \bar{\boldsymbol{z}}) + (\Delta\boldsymbol{z}_t^{nt} - \Delta\bar{\boldsymbol{z}}_t^{nt})\|_{H_t}.
\end{aligned}$$

Since for $\boldsymbol{z} \in \mathcal{T}$, $\|H_t\boldsymbol{z}\|_2 \geq \sqrt{m}\|\boldsymbol{z}\|_{H_t}$, (31) leads to

$$\begin{aligned}
\|\boldsymbol{x}_{t+1} - \bar{\boldsymbol{x}}\|_{H_t} &\leq \frac{1}{\sqrt{m}}\|H_t(\boldsymbol{z}_t - \bar{\boldsymbol{z}}) - H_t(\Delta\boldsymbol{z}_t^{nt} - \Delta\bar{\boldsymbol{z}}^{nt})\|_2 \\
&= \frac{1}{\sqrt{m}}\|H_t(\boldsymbol{z}_t - \bar{\boldsymbol{z}}) - (\boldsymbol{g}_t - \bar{\boldsymbol{g}})\|_2 \leq \frac{L_H}{2\sqrt{m}}\|\boldsymbol{z}_t - \bar{\boldsymbol{z}}\|_2^2,
\end{aligned} \tag{32}$$

where last inequality follows from Lipschitz-continuity of $\nabla^2 f(\boldsymbol{x})$. Since $\boldsymbol{z}_{t+1}, \bar{\boldsymbol{z}} \in \mathcal{T}$, we have

$$\|\boldsymbol{x}_{t+1} - \bar{\boldsymbol{x}}\|_{H_t} = \|\boldsymbol{z}_{t+1} - \bar{\boldsymbol{z}}\|_{H_t} \geq \sqrt{m}\|\boldsymbol{z}_{t+1} - \bar{\boldsymbol{z}}\|_2. \tag{33}$$

Finally, combining (33) with (32),

$$\|\boldsymbol{z}_{t+1} - \bar{\boldsymbol{z}}\|_2 \leq \frac{L_H}{2m}\|\boldsymbol{z}_t - \bar{\boldsymbol{z}}\|_2^2,$$

where quadratic convergence phase occurs when $\|\boldsymbol{z}_t - \bar{\boldsymbol{z}}\| < \sqrt{\frac{2m}{L_H}}$. $\qquad\square$

## 5 Numerical Experiments

In this section, we study the convergence behavior of Proximal Gradient method and Proximal Newton method on high-dimensional real data set with and without the CNSC condition. In particular, two loss functions — logistic loss $L(u, y) = \log(1 + \exp(-yu))$ and $\ell_2$-hinge loss $L(u, y) = \max(1 - yu, 0)^2$ — are used in (3) with $\ell_1$-regularization $h(\boldsymbol{x}) = \lambda\|\boldsymbol{x}\|_1$, where both losses are smooth but only logistic loss has strict convexity that implies the CNSC condition. For Proximal Newton method we employ an randomized coordinate descent algorithm to solve subproblem (10) as in [9]. Figure 5 shows their convergence results of objective value relative to the optimum on *rcv1.1k*, subset of a document classification data set with dimension $d = 10,192$ and number of samples $n = 1000$. From the figure one can clearly observe the linear convergence of Prox-GD and quadratic convergence of Prox-Newton on problem satisfying CNSC, contrasted to the qualitatively different behavior on problem without CNSC.

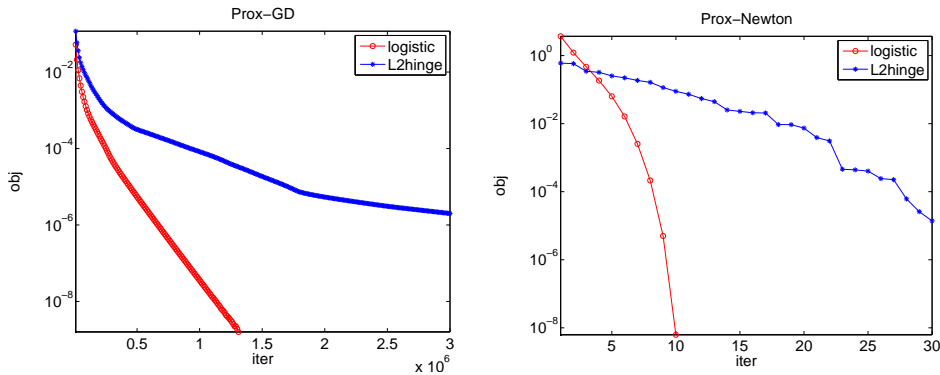

Figure 1: objective value (relative to optimum) of Proximal Gradient method (left) and Proximal Newton method (right) with logistic loss and $\ell_2$-hinge loss.

**Acknowledgement**

This research was supported by NSF grants CCF-1320746 and CCF-1117055. C.-J.H acknowledges support from an IBM PhD fellowship. P.R. acknowledges the support of ARO via W911NF-12-1-0390 and NSF via IIS-1149803, IIS-1320894, IIS-1447574, and DMS-1264033.

## Footnotes

[1] Note for many generalized linear models, the second derivative $L''(u, y)$ of loss function approaches 0 if $|u| \to \infty$. However, this could not happen as long as there is a penalty term $h(\boldsymbol{x})$ which goes to infinity if $\boldsymbol{x}$ diverges, which then serves as a finite constraint bound on $\boldsymbol{x}$.

[2] From our definition of decomposable norm, if a vector $v$ belongs to single subspace $M_j$, then $\|v\| = \|v\|_* = \|v\|_2$. The reason is: By the definition, if $v \in M_j$, then $v = c_j a_j$ for some $c_j > 0, a_j \in M_j, \|a_j\|_* = 1$, and it has decomposable norm $\|v\| = c_j$. However, we also have $\|v\|_* = \|c_j a_j\|_* = c_j\|a_j\|_* = c_j = \|v\|$. The norm equals to its dual norm only if it is $\ell_2$-norm.

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
