[Supplementary Material]

# Appendix: Constant Nullspace Strong Convexity and Fast Convergence of Proximal Methods under High-Dimensional Settings

**Ian E.H. Yen**       **Cho-Jui Hsieh**       **Pradeep Ravikumar**       **Inderjit Dhillon**
Department of Computer Science
University of Texas at Austin
{ianyen,cjhsieh,pradeepr,inderjit}@cs.utexas.edu

## 1   Proof for properties of proximal operations

The proximal operator $\mathbf{prox}(.)$ is defined as

$$\boldsymbol{x}_{t+1} = \mathbf{prox}(\boldsymbol{x}_{t+\frac{1}{2}}) = \operatorname*{arg\,min}_{\boldsymbol{x}}\ h(\boldsymbol{x}) + \frac{M}{2}\|\boldsymbol{x} - \boldsymbol{x}_{t+\frac{1}{2}}\|_2^2. \tag{1}$$

**Lemma 1.** *Define* $\Delta^P \boldsymbol{x} = \boldsymbol{x} - \mathbf{prox}(\boldsymbol{x})$, *the following properties hold for the proximal operation* (1).

1. $M\Delta^P \boldsymbol{x} \in \partial h(\mathbf{prox}(\boldsymbol{x}))$.

2. $\|\mathbf{prox}(\boldsymbol{x}_1) - \mathbf{prox}(\boldsymbol{x}_2)\|_2^2 \leq \|\boldsymbol{x}_1 - \boldsymbol{x}_2\|_2^2 - \|\Delta^P \boldsymbol{x}_1 - \Delta^P \boldsymbol{x}_2\|_2^2$.

*Proof.* The first property follows directly from the optimality condition of (1). The second property holds since for $M\Delta^P \boldsymbol{x}_1 \in \partial h(\mathbf{prox}(\boldsymbol{x}_1))$, $M\Delta^P \boldsymbol{x}_2 \in \partial h(\mathbf{prox}(\boldsymbol{x}_2))$ we have $\langle M\Delta^P \boldsymbol{x}_1 - M\Delta^P \boldsymbol{x}_2, \mathbf{prox}(\boldsymbol{x}_1) - \mathbf{prox}(\boldsymbol{x}_2)\rangle \geq 0$, and thus,

$$\|\boldsymbol{x}_1 - \boldsymbol{x}_2\|^2 = \|(\mathbf{prox}(\boldsymbol{x}_1) - \mathbf{prox}(\boldsymbol{x}_2)) + (\Delta^P \boldsymbol{x}_1 - \Delta^P \boldsymbol{x}_2)\|^2$$
$$\geq \|\mathbf{prox}(\boldsymbol{x}_1) - \mathbf{prox}(\boldsymbol{x}_2)\|^2 + \|\Delta^P \boldsymbol{x}_1 - \Delta^P \boldsymbol{x}_2\|^2,$$

which gives the second property. $\qquad\square$

The proximal operator $\mathbf{prox}_H(.)$ is defined for any PSD matrix $H$ as

$$\mathbf{prox}_H(\boldsymbol{x}) = \operatorname*{arg\,min}_{\boldsymbol{v}}\ h(\boldsymbol{v}) + \frac{1}{2}\|\boldsymbol{v} - \boldsymbol{x}\|_H^2. \tag{2}$$

**Lemma 2.** *Define* $\Delta^P \boldsymbol{x} = \boldsymbol{x} - \mathbf{prox}_H(\boldsymbol{x})$, *the following properties hold for the proximal operator:*

1. $H\Delta^P \boldsymbol{x} \in \partial h(\mathbf{prox}_H(\boldsymbol{x}))$.

2. $\|\mathbf{prox}_H(\boldsymbol{x}_1) - \mathbf{prox}_H(\boldsymbol{x}_2)\|_H^2 \leq \|\boldsymbol{x}_1 - \boldsymbol{x}_2\|_H^2$.

*Proof.* The first property follows directly from the optimality condition of (2). The second property holds since for $H\Delta^P \boldsymbol{x}_1 \in \partial h(\mathbf{prox}(\boldsymbol{x}_1))$, $H\Delta^P \boldsymbol{x}_2 \in \partial h(\mathbf{prox}(\boldsymbol{x}_2))$ we have $\langle H\Delta^P \boldsymbol{x}_1 - H\Delta^P \boldsymbol{x}_2, \mathbf{prox}(\boldsymbol{x}_1) - \mathbf{prox}(\boldsymbol{x}_2)\rangle \geq 0$, and thus,

$$\|\boldsymbol{x}_1 - \boldsymbol{x}_2\|_H^2 = \|(\mathbf{prox}_H(\boldsymbol{x}_1) - \mathbf{prox}_H(\boldsymbol{x}_2)) + (\Delta^P \boldsymbol{x}_1 - \Delta^P \boldsymbol{x}_2)\|_H^2$$
$$\geq \|\mathbf{prox}_H(\boldsymbol{x}_1) - \mathbf{prox}_H(\boldsymbol{x}_2)\|_H^2 + \|\Delta^P \boldsymbol{x}_1 - \Delta^P \boldsymbol{x}_2\|_H^2$$
$$\geq \|\mathbf{prox}_H(\boldsymbol{x}_1) - \mathbf{prox}_H(\boldsymbol{x}_2)\|_H^2,$$

where the second inequality follows from the PSD of $H$. $\qquad\square$

## 2 Proof of Lemma 3

**Lemma 3** (Optimal Set). *Let $\bar{\mathcal{E}}$ be the active set at optimal and $\bar{\mathcal{E}}^+ = \{j| \parallel \Pi_{\mathcal{M}_j}(\bar{\boldsymbol{\rho}})\parallel_* = \lambda\}$ be its augmented set (which is unique since $\bar{\boldsymbol{\rho}}$ is unique) such that $\Pi_{\mathcal{M}_j}(\bar{\boldsymbol{\rho}}) = \lambda\bar{\boldsymbol{a}}_j$, $j \in \bar{\mathcal{E}}^+$. The optimal solutions then form a polyhedral set*

$$\bar{\mathcal{X}} = \left\{\boldsymbol{x} \mid \Pi_{\mathcal{T}}(\boldsymbol{x}) = \bar{\boldsymbol{z}} \text{ and } \boldsymbol{x} \in \bar{\mathcal{O}}\right\}, \tag{3}$$

*where $\bar{\mathcal{O}} = \left\{\boldsymbol{x} \mid \boldsymbol{x} = \sum_{j\in\bar{\mathcal{E}}^+} c_j\bar{\boldsymbol{a}}_j, c_j \geq 0, j \in \bar{\mathcal{E}}^+\right\}$ is the set of $\boldsymbol{x}$ with $\bar{\boldsymbol{\rho}} \in \partial h(\boldsymbol{x})$.*

*Proof.* The optimality condition are $\boldsymbol{g}(\boldsymbol{x}) = \bar{\boldsymbol{g}}$ and $\bar{\boldsymbol{\rho}} \in \partial h(\boldsymbol{x})$ by Theorem 1. Since $\Pi_{\mathcal{T}}(\boldsymbol{x}) = \bar{\boldsymbol{z}}$, we have $\boldsymbol{g}(\boldsymbol{x}) = \bar{\boldsymbol{g}}$ already. Therefore, we only need to show that $\bar{\boldsymbol{\rho}} \in \partial h(\boldsymbol{x})$ iff $\boldsymbol{x} \in \bar{\mathcal{O}}$.

Suppose $\bar{\boldsymbol{\rho}} \in \partial h(\boldsymbol{x})$. Then for $j \notin \bar{\mathcal{E}}^+$, we know $\parallel\Pi_{\mathcal{M}_j}(\bar{\boldsymbol{\rho}})\parallel_* < 1$, which means $\Pi_{\mathcal{M}_j}(\boldsymbol{x}) = 0$, and for $j \in \bar{\mathcal{E}}^+$, we know $\Pi_{\mathcal{M}_j}(\bar{\boldsymbol{\rho}}) = \lambda\bar{\boldsymbol{a}}_j$, which means $\Pi_{\mathcal{M}_j}(\boldsymbol{x})$ can be $\boldsymbol{0}$ or $c_j\bar{\boldsymbol{a}}_j$ for some $c_j > 0$. Therefore, $\boldsymbol{x}$ must have the form $\boldsymbol{x} = \sum_{j\in\bar{\mathcal{E}}^+} c_j\bar{\boldsymbol{a}}_j, c_j \geq 0, j \in \bar{\mathcal{E}}^+$.

Now for the other direction, suppose $\boldsymbol{x} = \sum_{j\in\bar{\mathcal{E}}^+} c_j\bar{\boldsymbol{a}}_j, c_j \geq 0, j \in \bar{\mathcal{E}}^+$ and $\mathcal{E} \subseteq \bar{\mathcal{E}}^+$ is the set for which $c_j > 0, j \in \mathcal{E}$. Then since $\parallel\Pi_{\mathcal{M}_j}(\bar{\boldsymbol{\rho}})\parallel_* \leq 1, j \notin \mathcal{E}$ and for $j \in \mathcal{E} \subseteq \bar{\mathcal{E}}^+$ we have $\Pi_{\mathcal{M}_j}(\bar{\boldsymbol{\rho}}) = \lambda\bar{\boldsymbol{a}}_j$, we conclude that $\bar{\boldsymbol{\rho}} \in \partial h(\boldsymbol{x})$. $\qquad\square$

## 3 Proof of Lemma 5

**Lemma 5.** *Let $\bar{\mathcal{A}} = span(\bar{\boldsymbol{a}}_1, \bar{\boldsymbol{a}}_2 \ldots, \bar{\boldsymbol{a}}_{|\bar{\mathcal{E}}^+|})$. Suppose $\parallel\boldsymbol{x}\parallel \leq R$ and $\Pi_{\mathcal{M}_j}(\boldsymbol{x}) = \boldsymbol{0}$ for $j \notin \bar{\mathcal{E}}^+$. Then*

$$\lambda^2\parallel\boldsymbol{x} - \Pi_{\bar{\mathcal{A}}}(\boldsymbol{x})\parallel_2^2 \leq R^2\parallel\boldsymbol{\rho} - \bar{\boldsymbol{\rho}}\parallel_2^2,$$

*where $\boldsymbol{\rho} \in \partial h(\boldsymbol{x})$ and $\bar{\boldsymbol{\rho}}$ is as defined in Theorem 1.*

*Proof.* Since $\Pi_{\mathcal{M}_j}(\boldsymbol{x}) = \boldsymbol{0}$ for $j \notin \bar{\mathcal{E}}^+$, we have $\boldsymbol{x} = \sum_{j\in\bar{\mathcal{E}}^+} c_j\boldsymbol{a}_j$ for some $\boldsymbol{a}_j \in \mathcal{M}_j$. Then

$$\begin{aligned}
\parallel\boldsymbol{x} - \Pi_{\bar{\mathcal{A}}}(\boldsymbol{x})\parallel_2^2 &= \parallel \sum_{j\in\bar{\mathcal{E}}^+} c_j\boldsymbol{a}_j - \sum_{j\in\bar{\mathcal{E}}^+} c_j\langle\boldsymbol{a}_j, \bar{\boldsymbol{a}}_j\rangle\bar{\boldsymbol{a}}_j\parallel_2^2 \\
&= \sum_{j\in\bar{\mathcal{E}}^+} c_j^2\parallel\boldsymbol{a}_j - \langle\boldsymbol{a}_j, \bar{\boldsymbol{a}}_j\rangle\bar{\boldsymbol{a}}_j\parallel_2^2 \leq \sum_{j\in\bar{\mathcal{E}}^+} c_j^2\parallel\boldsymbol{a}_j - \bar{\boldsymbol{a}}_j\parallel_2^2.
\end{aligned}$$

Since $\Pi_{\mathcal{M}_j}(\boldsymbol{\rho}) = \lambda\boldsymbol{a}_j, \Pi_{\mathcal{M}_j}(\bar{\boldsymbol{\rho}}) = \lambda\bar{\boldsymbol{a}}_j$, we have

$$\parallel\boldsymbol{x} - \Pi_{\bar{\mathcal{A}}}(\boldsymbol{x})\parallel_2^2 \leq \frac{1}{\lambda^2} \sum_{j\in\bar{\mathcal{E}}^+} c_j^2\parallel\Pi_{\mathcal{M}_j}(\boldsymbol{\rho}) - \Pi_{\mathcal{M}_j}(\bar{\boldsymbol{\rho}})\parallel_2^2 \leq \frac{R^2}{\lambda^2}\parallel\boldsymbol{\rho} - \bar{\boldsymbol{\rho}}\parallel_2^2$$

as claimed. $\qquad\square$

## 4 Proof of Lemma 6

**Lemma 6** (Optimality Condition). *For any matrix $H$ satisfying CNSC-$\mathcal{T}$, the update*

$$\Delta\boldsymbol{x} = \underset{\boldsymbol{d}}{argmin} \quad h(\boldsymbol{x} + \boldsymbol{d}) + \boldsymbol{g}(\boldsymbol{x})^T\boldsymbol{d} + \frac{1}{2}\parallel\boldsymbol{d}\parallel_H^2 \tag{4}$$

*has*

$$F(\boldsymbol{x} + t\Delta\boldsymbol{x}) - F(\boldsymbol{x}) \leq -t\parallel\Delta\boldsymbol{z}\parallel_H^2 + O(t^2), \tag{5}$$

*where $\Delta\boldsymbol{z} = \Pi_{\mathcal{T}}(\Delta\boldsymbol{x})$. Furthermore, if $\boldsymbol{x}$ is an optimal solution, $\Delta\boldsymbol{x} = \boldsymbol{0}$ satisfies* (4).

*Proof.* By smoothness of $f(\boldsymbol{x})$ and convexity of $h(\boldsymbol{x})$, we have

$$\begin{aligned}
F(\boldsymbol{x} + t\Delta\boldsymbol{x}) - F(\boldsymbol{x}) &= h(\boldsymbol{x} + t\Delta\boldsymbol{x}) - h(\boldsymbol{x}) + f(\boldsymbol{x} + t\Delta\boldsymbol{x}) - f(\boldsymbol{x}) \\
&\leq t(h(\boldsymbol{x} + \Delta\boldsymbol{x}) - h(\boldsymbol{x})) + \boldsymbol{g}(\boldsymbol{x})^T(t\Delta\boldsymbol{x}) + \mathcal{O}(t^2).
\end{aligned} \tag{6}$$

Then we try to bound the descent amount predicted by gradient $t(h(\boldsymbol{x} + \Delta \boldsymbol{x}) - h(\boldsymbol{x}) + \boldsymbol{g}(\boldsymbol{x})^T \Delta \boldsymbol{x})$. Since $\Delta \boldsymbol{x}$ is optimal solution of (4), we have

$$h(\boldsymbol{x} + \Delta \boldsymbol{x}) + \boldsymbol{g}(\boldsymbol{x})^T \Delta \boldsymbol{x} + \frac{1}{2} \|\Delta \boldsymbol{x}\|_H^2$$

$$\leq h(\boldsymbol{x} + t\Delta \boldsymbol{x}) + \boldsymbol{g}(\boldsymbol{x})^T (t\Delta \boldsymbol{x}) + \frac{1}{2} \|t\Delta \boldsymbol{x}\|_H^2 \tag{7}$$

$$\leq th(\boldsymbol{x} + \Delta \boldsymbol{x}) + (1-t)h(\boldsymbol{x}) + \boldsymbol{g}(\boldsymbol{x})^T (t\Delta \boldsymbol{x}) + \frac{1}{2} \|t\Delta \boldsymbol{x}\|_H^2,$$

which implies

$$(1-t)(h(\boldsymbol{x} + \Delta \boldsymbol{x}) - h(\boldsymbol{x})) + (1-t)\boldsymbol{g}(\boldsymbol{x})^T \Delta \boldsymbol{x} + \frac{1-t^2}{2} \|\Delta \boldsymbol{x}\|_H^2 \leq 0, \tag{8}$$

and therfore,

$$(h(\boldsymbol{x} + \Delta \boldsymbol{x}) - h(\boldsymbol{x})) + \boldsymbol{g}(\boldsymbol{x})^T \Delta \boldsymbol{x} \leq -\frac{1+t}{2} \|\Delta \boldsymbol{x}\|_H^2 = -\frac{1+t}{2} \|\Delta \boldsymbol{z}\|_H^2, \tag{9}$$

where $\Delta \boldsymbol{z} = \Pi_{\mathcal{T}}(\Delta \boldsymbol{x})$ and last inequality follows from CNSC-$\mathcal{T}$ of $H$. Let $t \to 1$ and combine (9) and (6), we obtain

$$F(\boldsymbol{x} + t\Delta \boldsymbol{x}) - F(\boldsymbol{x}) \leq -t\|\Delta \boldsymbol{z}\|_H^2 + \mathcal{O}(t^2), \tag{10}$$

which shows $\Delta \boldsymbol{x}$ obtained from (4) is a descent direction if $\Delta \boldsymbol{z} \neq \boldsymbol{0}$.

Now suppose $\boldsymbol{x}$ is an optimal solution of $F(\boldsymbol{x})$. Then the $\Delta \boldsymbol{x}$ defined in (4) cannot be a descent direction, which means $\Delta \boldsymbol{z}$ must be $\boldsymbol{0}$. However, since $f(\boldsymbol{x})$ and $H$ satisfy CNSC-$\mathcal{T}$, when $\Delta \boldsymbol{z} = \boldsymbol{0}$, (4) reduced to

$$\Delta \boldsymbol{x} = \underset{\Delta \boldsymbol{y} \in \mathcal{T}^\perp}{argmin} \quad h(\boldsymbol{x} + \Delta \boldsymbol{y}). \tag{11}$$

$\Delta \boldsymbol{x} = \boldsymbol{0}$ satisfies (11) since $\boldsymbol{x} = \boldsymbol{y} + \boldsymbol{z}$ is already a minimum of $h(\boldsymbol{x}) + f(\boldsymbol{x})$, while $f(\boldsymbol{x})$ does not depend on $\boldsymbol{y}$, where $\boldsymbol{y} = \Pi_{\mathcal{T}^\perp}(\boldsymbol{x})$. □

## 5 Proof of Lemma 7

**Lemma 7.** *Suppose $h(\boldsymbol{x})$ and $f(\boldsymbol{x})$ are Lipchitz-continuous with Lipchitz constants $L_h$ and $L_f$. In quadratic convergence phase (defined in Theorem 3), Proximal Newton Method has*

$$F(\boldsymbol{x}_t) - F(\bar{\boldsymbol{x}}) \leq L\|\boldsymbol{z}_t - \bar{\boldsymbol{z}}\|, \tag{12}$$

*where $L = \max\{L_h, L_f\}$ and $\boldsymbol{z}_t = \Pi_{\mathcal{T}}(\boldsymbol{x}_t)$, $\bar{\boldsymbol{z}} = \Pi_{\mathcal{T}}(\bar{\boldsymbol{x}})$.*

*Proof.* LWe prove (12) by showing that $|f(\boldsymbol{z}_1) - f(\boldsymbol{z}_2)| \leq L_f\|\boldsymbol{z}_1 - \boldsymbol{z}_2\|$ and $|h(\boldsymbol{z}_1 + \hat{\boldsymbol{y}}(\boldsymbol{z}_1)) - h(\boldsymbol{z}_2 + \hat{\boldsymbol{y}}(\boldsymbol{z}_2))| \leq L_h\|\boldsymbol{z}_1 - \boldsymbol{z}_2\|$ for any $\boldsymbol{z}_1 \in \mathcal{T}, \boldsymbol{z}_2 \in \mathcal{T}$. Since $f(\boldsymbol{z})$ does not depend on the null-component $\boldsymbol{y}$, the first inequality holds directly from the Lipchitz-continuity of $f(\boldsymbol{z})$. The second inequality holds since

$$h(\boldsymbol{z}_1 + \hat{\boldsymbol{y}}(\boldsymbol{z}_1)) \leq h(\boldsymbol{z}_1 + \hat{\boldsymbol{y}}(\boldsymbol{z}_2)) \leq h(\boldsymbol{z}_2 + \hat{\boldsymbol{y}}(\boldsymbol{z}_2)) + L_h\|\boldsymbol{z}_1 - \boldsymbol{z}_2\|$$

and

$$h(\boldsymbol{z}_2 + \hat{\boldsymbol{y}}(\boldsymbol{z}_2)) \leq h(\boldsymbol{z}_2 + \hat{\boldsymbol{y}}(\boldsymbol{z}_1)) \leq h(\boldsymbol{z}_1 + \hat{\boldsymbol{y}}(\boldsymbol{z}_1)) + L_h\|\boldsymbol{z}_1 - \boldsymbol{z}_2\|$$

by the definition of $\hat{\boldsymbol{y}}(\boldsymbol{z}_1), \hat{\boldsymbol{y}}(\boldsymbol{z}_2)$ and Lipchitz-continuity of $h(\boldsymbol{x})$. □