[Reviews · NeurIPS 2014]

Submitted by Assigned_Reviewer_19

This paper investigate fast convergence properties of proximal gradient method and proximal Newton method under the assumption of Constant Nullspace Strong Convexity (CNSC). The problem of interest is to minimize the sum of two convex functions f(x)+h(x), where f is twice differentiable (smooth) and h can be non-smooth but admits a simple proximal mapping. Under the CNSC assumption on f and assuming h has the form of decomposable norm, this paper showed global geometric convergence of the proximal gradient method, and local quadratic convergence of the proximal Newton method.

Writing of this paper is very clear. My main concern is that the CNSC condition is too strong, or correspondingly, the number m in Assumption 1 can be too small to have meaningful linear convergence bounds in practice. This may severely limits its applicability in explaining the behavior of proximal gradient and Newton methods.
As an example, for logistic regression, the second-order derivative L"(u,y) can be arbitrarily small if as u become large, so the bound m_L can be arbitrarily small.

For least square problems, the bound lambda_min(A^T*A), denoting the smallest non-zero eigenvalue of A^T*A, can be very small even though it is fixed for given data A, which may not explain the fast convergence of proximal gradient/Newton method when applied to, for example, Lasso for sparsity recovery problems. When the solution is sparse, the local strong convexity is characterized by restricted eigenvalues with sparse support, which can be much bigger than lambda_min(A^T*A). In such a context, the local analysis in references [5,7,17] provide more meaningful bounds.

Of course, one can argue that the results in this paper do not depend on additional assumptions like restricted eigenvalue conditions or alike. However, in this case, a sublinear convergence rate may give more meaningful prediction of the algorithmic behavior for a finite precision, than linear rates with tiny exponents. Theoretical and empirical comparisons in this regard are needed to further reveal the relevance of this work.
Summary: This paper shows fast convergence of proximal gradient and Newton methods under the Constant Nullspace Strong Convexity assumption. The fast rates obtained do not depends on additional assumptions such as restricted eigenvalue conditions, but can be very weak to predict realistic behavior of the algorithms. Further remarks and analysis regarding relevance of CNSC in real or random data can be more helpful.

Submitted by Assigned_Reviewer_32

This paper proposes a new sufficient condition for guaranteeing the convergence of proximal gradient method and proximal Newton method. The results are interesting and important. But the presentation is not very clear. My detailed comments are as follows.

Line 128: to better motivate the new condition, the authors should provide more detailed examples. For example, how does logistic regression satisfy the new condition? What is the corresponding T here?

Theorem 2: it should be commented that the error bound constant \theta(\bar{X}) is not known in practice. So this result is only of theoretical use.

Section 4 in page 7: again, what is T in this part? Since this is crucial to the analysis, it is better to give more explanations on T. For example, in Section 5, the authors considered logistic loss function again. So what is T here for this function?
Summary: This paper has interesting results. But the presentation in some places is not clear.

Submitted by Assigned_Reviewer_40

This paper studies constant nullspace strong convexity, strongly convex only over a constant subspace, that can result in a linear convergence rate for the proximal gradient method and quadratic convergence rate for the proximal Newton method, even if the objective function is not strongly convex on the whole space. This work is an important extension of recent convergence rate analysis on sparse and low-rank minimization problems, where the objective functions are usually l1 norm or nuclear norm and the linear operators in the constraint usually have some nullspace property. The authors also extend the analysis to general decomposable norms.

However, the paper has the following issues:
1. The proximal Newton method does not seem practical in reality. The subproblem (10) is usually not easily solvable, unless h(x) is a quadratic function. That all subproblems should be easily solvable is a basic requirement on a practical algorithm. Otherwise, may I simply assume that (1) is easily solvable, whatever it is? Indeed, the authors have used randomized coordinate descent to solve (10), which hardly has any guarantee on accuracy of the solution to (10). So the subsequent quadratic convergence does not make sense at all. This algorithm is also awkward as it requires an extra linear search. And the authors do not specify when to stop line search (only wrote "sufficient decrease"). So I don't think analysis on the proximal Newton method adds much contribution.
2. The authors overclaim the contribution a bit. Actually, this paper only analyzes the case of h(x) being a decomposable norm, not a general convex function. So the title, abstract and introduction should all be fixed.
3. There may be some errors in the proofs that need clarification from the authors:
a. In (22), it is unclear why the first inequality holds.
b. In the third line below (22), it is unclear why the l2-norms of a_j and \bar{a}_j are both 1. Note that in the definition decomposable norm, the dual norms of a_j and \bar{a}_j are both 1.
4. In the second line below (29), "\epsilon" and "L\epsilon" should switch. In experiments, where is 'x' in the loss functions? And why the minimal objective function values are both zeros? Finally, the authors assume that f(x) is twice continuous differentiable but I don't think that the hinge loss satisfies this condition.
5. In Fig. 1, I don't think that it is obvious that the upper curve corresponds to linear convergence. And the convergence is surprisingly slow -- at the order of millions of iterations! Similarly, in Fig. 2 that the upper curve is in quadratic convergence is not obvious either.
Summary: The authors proposed an interesting theory on the linear convergence of some algorithms for sparse and low-rank minimization problems, although the objective functions are not strongly convex. However, the proximal Newton method does not seem practical in reality. So I don't think analysis on the proximal Newton method adds much contribution. The experiments are a bit weak.
Author Feedback
Author rebuttal: We are thankful to all reviewers for their careful and constructive comments.

Reviewer_19:

1. Regarding concerns that the constant "m" in CNSC condition (6) could be too small in practice to yield useful result.
"As an example, for logistic regression, the second-order derivative L"(u,y) can be arbitrarily small if as u become large, the bound m_L can be arbitrarily small."

For proximal algorithms considered here, u is actually bounded for all iterates x_t, t=0,1,2,.... if h(x)= \lambda \|x\|, and \lambda> 0. Since the algorithms guarantee descent of each iteration, we have \lambda\|x_t\| <= \lambda\|x_t\|+f(x_t) <= f(0) and thus u = < a_i, x_t > <= \|a_i\|_* \|x_t\| <= \frac{ \|a_i\|_* f(0) }{ lambda }. Therefore, since u is bounded, L"(u,y) is bounded away from 0.

On the other hand, the smallest positive eigenvalue eig_min(A^T*A) does depend on the data. In our experiment, the data set "rcv-1k" (d=10,192, n=1000) has eig_min(A^T*A)=0.0034.

Reviewer_32:

1. Regarding clarifications on how CNSC condition applies in the problems, including for logistic regression.

For any loss minimization problem as in eqn. (3) in the paper (including for logistic regression), T^{\perp} should be the nullspace of the *data matrix A*, and T is its orthogonal space. Since H=A'DA, we have Hx=0 if x\in T^{\perp}, and x'Hx >= m||x||^2 if x \in T, where m=m_L*lambda_min, m_L is lower bound of L''(u,y) and lambda_min is smallest positive eigenvalue of A’A.

We will add further clarifications and examples in the final version.

Reviewer_40:

1. Regarding concerns about practical efficiency of Prox-Newton method, given that sub-problem (10) is not easily solvable.

Prox-Newton has been previously investigated by several recent papers [1,2,8,9,21] and has been shown to be practically successful in problems for which subproblem (10) is more tractable than the original problem(due to Hessian having special structure), such as Sparse Inverse Covariance Estimation [2] and L1-regularized logistic regression [9], where a coordinate descent update on (10) is cheaper than that directly applied on (1). Indeed, in practice sub-problem (10) is not solved exactly, so an analysis assuming inexact solutions would definitely be interesting. Our analysis of exact version, however, serves as an important milestone.

2. Regarding concerns with overclaiming from decomposable norm to general convex h(x).

While the assumption of decomposable h(x) did not appear in the title, it appears in the Abstract (Line 29-30), and in the introduction (Lines 71-73). Note moreover that the proof of Prox-GD assumes a decomposable norm, but the proof of Prox-Newton only assumes h(x) being convex, Lipschitz-continuous. We will add clarifications in the final version.

3. Regarding places in proof that require clarification: (a). 1st inequality of (22) and (b). last equality in Line 306.

From our definition of decomposable norm, if a vector v belongs to single subspace M_j, then \|v\|=\|v\|_* =\|v\|_2. The reason is:

By the definition, if v \in M_j, then v=c_j a_j for some c_j>0, a_j \in M_j, \|a_j\|_*=1, and it has decomposable norm \|v\|=c_j. However, we also have \|v\|_* = \| c_j a_j\|_* = c_j \|a_j\|_* = c_j = \|v\|. The norm equals to its dual norm only if \|v\|=\|v\|_* =\|v\|_2.

This explains the change of norm in (22) and Line 306. We will add further clarifications in the final version.

4. Regarding the experiment section:

i. "In experiments, where is 'x' in the loss functions?"

The expression of objective follows (3); given the loss function L(u,y), the objective (following (3)) is given by f(x) = \sum_i L(a_i^Tx, y_i).

ii. "why the minimal objective function values are both zeros?"

The y-axis shows relative objective function value: (F(x)-F*)/F*, where F* is the reference objective value, which should be the minimum regarding the chosen loss function; In experiment, we run the solver till a strict stopping condition (||gradient||<1e-16) to get the reference F*.

iii. "the authors assume that f(x) is twice continuous differentiable but I don't think that the hinge loss satisfies this condition."

The hinge loss satisfies neither CNSC nor twice-differentiability, and its curves are indeed not linearly convergent (for Prox-GD) nor quadratically convergent (for Prox-Newton). Indeed, we plotted its curves to contrast the difference in convergence behavior between the logistic loss that satisfies our condition and the L2-Hinge loss that does not, as we noted in Lines 406-410. We will add further clarifications to the experiments section on this point.